# Effects of Climate Conditions before Harvest Date on Edamame Metabolome

**DOI:** 10.3390/plants13010087

**Published:** 2023-12-27

**Authors:** Akira Oikawa, Katsutaka Takeuchi, Kei Morita, Yamato Horibe, Ryosuke Sasaki, Hideki Murayama

**Affiliations:** 1Graduate School of Agriculture, Kyoto University, Kitashirakawaoiwake-cho, Sakyo-ku, Kyoto 606-8502, Japan; 2Faculty of Agriculture, Yamagata University, Wakaba-machi 1-23, Tsuruoka 997-8555, Japanmhideki@tds1.tr.yamagata-u.ac.jp (H.M.); 3RIKEN Center for Sustainable Resource Science, Suehiro-cho 1-7-22, Tsurumi-ku, Yokohama 230-0045, Japan; ryosuke.sasaki@riken.jp

**Keywords:** edamame, climate condition, annual fluctuation, metabolomics

## Abstract

Edamame is a green soybean that is rich in nutrients. Boiled edamame has been traditionally used for food in the East Asia region. It was known among farmers that conditions, such as temperature and climate on the day of harvest, affect the quality of edamame. Large-scale farmers harvest edamame on multiple days in the same year; however, the quality of edamame varies from day to day due to variations in climate conditions. In this study, we harvested edamame over several days between 2013 and 2018, obtained the climate conditions on the harvest date, and performed metabolome analysis using capillary electrophoresis mass spectrometry. To clarify the correlation between climate conditions before the harvest date and edamame components, comparative analyses of the obtained meteorological and metabolomic data were conducted. We found positive and negative correlations between the sunshine duration and average temperature, and the amounts of some edamame components. Furthermore, correlations were observed between the annual fluctuations in climate conditions and edamame components. Our findings suggest that the climate conditions before the date of harvesting are closely related to edamame quality.

## 1. Introduction

Edamame, the immature seed of the soybean (*Glycine max* L. *Merr.*), is mainly eaten in Japan and other East Asian countries. Soybeans, which are fully ripened seeds, are cultivated as one of the major crops around the world. Because the seeds are rich in protein and lipids, they have a wide range of uses, including food, edible oil, and livestock feed. Similar to soybeans, edamame is rich in protein, sugar, fat, vitamin C, and vitamin B_1_; thus, it is a popular food, as it has high nutritional value and taste [1]. Many studies have been conducted on the conditions for harvesting high-quality edamame efficiently by analyzing information such as the variety, planting date, harvest date, edamame color, and pod weight using artificial neural network analysis [2] by using digital imaging technology and computer vision algorithms to characterize the major traits of edamame’s shoot architecture [3], or by applying a spectroscopy-based machine learning technique [4]. In recent years, ethyl methanesulfonate-based breeding improvements and genome-wide-association study-based searches for useful genes have been performed, and these obtained unique mutants flowered considerably earlier than the wild type and identified single nucleotide polymorphisms (SNPs) related to the amounts of sucrose and Ala, respectively [5,6]. Previous studies reported the health functionality of edamame, especially its effects on obesity and diabetes [7,8]. In addition, practical research has being conducted, such as a previous comprehensive analysis of the changes in edamame composition due to storage conditions using actual distribution conditions [9].

Differences in the microclimate, caused by geographical conditions or artificial agricultural techniques, affect the quality of agricultural products. In grape-producing regions, such as those in Europe, there are differences in the grape quality from field to field, and this difference is thought to be due to various environmental factors, including the microclimate called terroir [10]. Previous studies reported that the temperature of the field environment affects the coloration of grapes and that excessive sunlight has a negative effect on the amount of flavonoids and flavor-related components [11,12]. Metabolome analysis has also revealed that sunlight causes changes in the amounts of other metabolites in grapes [13,14]. It has also been reported that climate conditions affect the volatile amount of grapes and are related to the quality of wine [15,16]. In addition, there are many reports of a relationship between the quality of agricultural products and meteorological conditions, such as sunshine duration, rainfall, and temperature, during cultivation, including research on abiotic stress responses. For example, in tomatoes, the level of antioxidant metabolites, such as ascorbic acid and lycopene, fluctuate because of temperature and sunlight [17], and the amount of sugars and carotenoids was changed using artificial fog [18]. In rice, starch accumulation was inhibited, and amino acid accumulation was induced in grains during ripening by high temperatures [19,20]. Climate conditions have been reported to affect the quality of agricultural products such as asparagus [21], common beans [22], cherries [23], and pomegranates [24]. However, most of these studies used agricultural products that had been exposed to the same climate conditions throughout the study period and used agricultural materials such as shade or mulch. Few studies have investigated the relationship between the differences in climate conditions for several days before harvesting and the quality of agricultural products.

In this study, the same variety of edamame was harvested under different climate conditions. A previous study reported that the cumulative amount of climate over two or more days is important for physiological phenomena such as flowering and the germination of plants. We hypothesized that the climate conditions just before harvest would affect the quality of edamame soybeans. We collected climate information, such as the sunshine duration (SD), averaged temperature (AT), and rainfall (RT) before the day of harvest, as well as the metabolomic information contained in harvested edamame using capillary electrophoresis mass spectrometry (CE-MS) over a 6-year period. We also investigated the relationship between the cumulative values of sunshine duration, average temperature, and rainfall from nine days before harvest to the day before harvest, and edamame quality.

## 2. Results

We found a correlation between the climate conditions before the harvest date and the amounts of several compounds contained in edamame. Furthermore, annual fluctuations in the climate conditions affected the edamame metabolome depending on the year of harvest.

### 2.1. Climate Conditions before Harvest Days

#### 2.1.1. Sunshine Duration, Average Temperature, Rainfall, Average Wind Speed, and Temperature Difference on the Day before Harvest

The temperature was observed using a data logger installed near an edamame field. For the sunshine duration, rainfall, and average wind speed (AWS), we referred to meteorological data from Tsuruoka City, where the field is located. We calculated the average temperature and temperature difference (TD). TD means the difference between the highest and lowest temperature on the harvest day. Figure 1 shows climate data for 108 days between 2013 and 2018. It was found that there was a large daily variation in the duration of sunshine every year, and there were days with no sunshine and days with more than 12 h of sunshine (Figure 1A). The average temperature was between 20 °C and 30 °C; however, the average temperature varied by more than 5 °C, even within the same year (Figure 1B). During the harvest period, there were many days without rain, but there were some days when the rainfall exceeded 100 mm due to torrential rain (Figure 1C). The average wind speed also varied widely, and there were several days with strong winds that were probably caused by typhoons (Figure 1D). Differences in temperature also varied greatly, even within the same year, with some days having a difference of more than 10 °C and some days having a difference of 2 °C (Figure 1E).

#### 2.1.2. Cumulative Sunshine Duration, Average Temperature, and Rainfall in the Days before Harvest

The cumulative sunshine hours, average temperature, and rainfall for 3, 5, 7, and 9 days from the day before harvest are summarized in Appendix A. Compared to the data on the day before harvest, the accumulated values show the difference in climate conditions on the day of harvest. The upward slope of daylight hours for each year indicates that the climate has become more suitable for edamame growing during the harvest period, especially in 2014, 2017, and 2018. The average temperature data shows that the temperature decreased during the harvest period in 2015. Since there is a correlation between fields becoming moist due to rain and the cumulative amount of rainfall, it is assumed that the soil moisture amount was relatively high in 2016 and 2018.

### 2.2. Metabolome of Harvested Edamame

Edamame of the same variety were harvested at the same time (7 to 8 a.m.) every year using the same method in the same field. Boiling the edamame immediately after harvest prevented the loss of freshness and made the edamame ready for consumption. After each sample of these boiled edamame was extracted and pre-treated using conventional methods, metabolome analysis was performed using CE-MS. Fifty-eight types of ionic compounds were quantified in edamame produced between 2013 and 2018 (Appendix A). These included amino acids, such as alanine and glutamic acid; organic acids, such as citric acid and fumaric acid; and nucleotides, such as AMP and GDP. Alanine, the amino acid that is most abundant in boiled edamame, showed a more than double amount difference depending on the harvest date, even in the same year (Figure 2). Similarly, the citric acid concentration more than doubled within the same year; however, the diurnal variation pattern was different from that of alanine (Appendix A). These results indicate that the amount of compounds contained in edamame differed depending on the harvest date. Moreover, the effect was different depending on the variety of compounds.

### 2.3. Correlation Analysis between Climate Conditions up to the Day before Harvest and Compound Concentrations in Harvested Edamame

#### 2.3.1. Correlation Analysis between Climate Conditions and Edamame Metabolome

Correlation analysis was performed using six years of meteorological and metabolomic data (Figure 3). We found correlations between the metabolites and between climate conditions and metabolite amounts. Amounts of some amino acids in edamame were correlated with the average temperature and sunshine duration. A strong correlation was observed between lysine (Lys) and the average temperature, whereas inverse correlations with the sunshine duration and average temperature were observed for some organic acids and nucleotides. There was no clear correlation among rainfall, average wind speed, temperature difference, and the amount of each compound.

#### 2.3.2. Compounds That Were Correlated with Sunshine Duration and Average Temperature

Table 1 and Table 2 show compounds that showed a correlation or inverse correlation with the sunshine duration and average temperature, respectively. Interestingly, these values showed different trends depending on the climate conditions and the amount of correlation. For example, some amino acids were found to be correlated with the amount of sunlight accumulated 3 or 5 days before harvest, whereas a high correlation with the average temperature was found with the amount accumulated 1 or 9 days before harvest. On the other hand, regarding the inverse correlation coefficient, a high correlation was observed with the amount accumulated over 5 days after harvest. For example, a positive correlation to Tyr was acquired by an accumulated sunshine duration for 5 days; however, the highest positive correlation of Tyr with the average temperature required accumulation of the average temperature for 9 days. These data show that climate conditions before the harvest date change the amount of some of the compounds in edamame; however, there seems to be a difference in how many days the seed must pass under a given climate condition before a response is observed.

### 2.4. Annual Fluctuation between Meteorological Conditions and Edamame Metabolome

#### 2.4.1. Annual Fluctuation of Climate Conditions

We confirmed the annual changes in climate conditions over the 6-year period. Sunshine duration is a meteorological parameter with clear annual fluctuations, and it was higher in 2013 and 2016 (Appendix A). A particularly clear difference was observed in the accumulated sunshine duration during the five days before harvest, with a difference of about 30 h when there was more sunlight, compared to 2014 when there was less (Figure 4). Similar annual fluctuations were observed in the average temperature (Appendix A). The accumulated average temperature during the 5 days before harvesting also showed high values in 2013 and 2016, as did the accumulated sunshine durations (Figure 4). On the other hand, no clear annual fluctuations were observed in rainfall, average wind speed, and temperature difference (Appendix A).

#### 2.4.2. Annual Fluctuation of Edamame Metabolome

Similar to climate conditions, the amounts of metabolites in edamame also showed annual variations. To clarify the correlation between the edamame metabolome and the climate conditions before the harvest date, we performed principal component analysis (PCA) (Figure 5). The 2013 and 2016 clusters and other clusters were separated along PC1 (Figure 5A). In PC2, no difference between years was observed (Appendix A), but in PC3, the data for 2018 were separated in a positive direction from those of others (Figure 5A). From the loading plot, it was found that sunshine duration, average temperature, and the amounts of some amino acids, such as Ala, Tyr, and Lys, had positive effects on cluster separation in 2013 and 2016 from others, while citric acid, isocitric acid, GDP, ADP, etc., were found to have a negative impact on the clusters of 2013 and 2016 (Figure 5B). Rainfall, along with guanosine and cytidine, influenced the positive direction of PC3 (Figure 5B). The annual fluctuations in sunshine duration and average temperature matched those expected from the graphs of average values (Figure 4 and Appendix A); however, the rainfall contribution to variance was only isolated by PCA. The positive correlation between the sunlight duration and average temperature for the number of amino acids, such as Ala, and the negative correlation for the number of organic acids, such as citric acid, and nucleotides, such as GDP, agreed with the results of the correlation analysis (Table 1 and Table 2). The 2018 cluster suggested correlations with rainfall and the amount of nucleosides such as guanosine and cytidine. Furthermore, hierarchical cluster analysis (HCA) was performed to clarify the relationship between weather conditions and the content of compounds in edamame (Figure 6). The cluster at the bottom of this figure (from Phe to SD_3) contained multiple amino acids, including Lys and tyrosine (Tyr), as well as the average temperature, sunshine duration, and accumulated values (Figure 6). This result indicated a strong association between these metabolites and climate conditions. The large central cluster in Figure 5 (from RF_1 to HydroxyPro) included rainfall values and metabolites such as GABA and choline. Many organic acids and nucleotides, such as citric acid and GDP, were identified in clusters separated in the upper group. These metabolites did not show an association with any climate conditions. However, the metabolites clustered in this group indicated inverse correlations to the sunshine duration and average temperature (Table 1 and Table 2, Figure 5). Figure 7 shows the variation in the amounts of the four representative metabolites in the lower and upper clusters with respect to the year of harvest. Compared to the annual variations in sunshine duration and the average temperature in Figure 4, these characteristic metabolites showed similar or opposite annual variations to these climatic conditions (Figure 7), as shown in PCA (Figure 5).

## 3. Discussion

Plants display a higher metabolic plasticity in response to the environment since, unlike animals, they cannot take shelter. Therefore, plants biosynthesize various metabolites in response to stress. It has been reported that environmental conditions, such as temperature and light intensity, affect the amount of metabolites in plants [25,26,27,28,29]. Most reports have focused on secondary metabolites, such as flavonoids and terpenoids; however, the effects of environmental conditions on the number of primary metabolites, such as sugars and organic acids, that are important for the quality of agricultural products, have also been investigated. The effects of temperature, ultraviolet radiation, and sunlight duration on fruit quality and composition were studied using nine strawberry varieties grown at three different altitudes [30]. They reported a correlation between the phenylpropanoid amount and sunshine duration. Moreover, the pH value, which is thought to be caused by the sugar amount and organic acid amount in fruit, changes depending on the light intensity and temperature. A previous study reported that differences in light-shielding conditions in a greenhouse and a field affected the amounts of sugars, amino acids, and organic acids contained in maize [31]. However, these previous studies compared climate conditions and metabolite amounts over a long period before harvest and did not investigate the more detailed changes in climate conditions. In the present study, we focused on the weather conditions for one to nine days before harvesting. We found that minute changes in climate conditions before the harvest date affected the amount of metabolites related to the quality of agricultural products. Specifically, an accumulated sunshine duration and average temperature in the several days before harvest were positively correlated with some amino acids and negatively correlated with organic acid and the nucleotide amount in edamame (Figure 3). This may be true not only for edamame but also for other agricultural products.

The quality of agricultural products varies depending on the year of harvest. In the case of wine, the year of production is important in addition to the brewery and grape variety. However, there are few studies on the annual changes in low-molecular-weight compounds contained in agricultural products. A study that examined the leaf area, leaf dry weight, plant height, and chlorophyll amount in 12 shortgrass species over a 4-year period found that these traits had a large influence depending on the year, although the detailed reason was not clarified [32]. It is also known that the sterol composition contained in dandelion leaves changes depending on the amount of sunlight and temperature [33]. A previous study aimed to evaluate the use of organic fertilizers by examining the amount of δ^15^N contained in tea leaves over multiple years [34]. Our comparative analysis of meteorological conditions and edamame metabolites over a 6-year period suggested that the sunshine duration and average temperature have a significant influence on the amount of some compounds in edamame (Figure 4, Figure 5 and Figure 6).

Amino acids are abundant in edamame and are involved in its flavor [9,35,36]. We found that the amino acid amount in edamame was affected by the sunlight duration and average temperature up to the day before harvest. Proline (Pro) has been reported to accumulate in plants in response to stresses, such as heat and drought, and it is also known that Pro application confers tolerance to these stresses in plants [37,38,39,40]. In this study, the amount of Pro in edamame also increased with increases in the sunlight duration and average temperature (Table 1 and Table 2). A long sunshine duration and high average temperatures can cause heat stress and drought stress to edamame plants. The edamame used in this study may also have been subjected to these stresses, depending on the conditions of sunlight duration and average temperature. Similar to Pro, Lys and Tyr were also found to be correlated with the sunshine duration and average temperature. The fluctuation of the levels of amino acids, including Lys and Tyr, in plants treated with abiotic stresses has been reported in metabolomic studies [41,42]. Lys-derived alkaloids and Tyr-derived flavonoids are known to be biosynthesized in leguminous plants in response to various stresses [43,44]. Similar to Pro, their role in the stress response is not clear, but since these amino acids serve as raw materials for various secondary metabolites, they may be biosynthesized as part of edamame’s response to stress caused by sunlight and temperature.

The amounts of organic acids and nucleotides showed a negative correlation with the accumulated sunshine duration and average temperature in the few days before harvest (Figure 5 and Figure 7). Sugar and citric acid amounts were reduced in tanned mandarin fruits [45]. In this study, the observed decrease in these metabolites correlates with higher sunlight exposure and a larger temperature difference. In soybean leaves, the amounts of citric acid and malic acid decreased significantly under high-temperature conditions [46]. In addition, in tomato fruits, citric acid and malic acid increased and decreased, respectively, at high temperatures; however, these showed different responses to nitrogen fertilization [47]. In grapes, the amount ratio of tartaric acid and malic acid changed depending on the temperature conditions [48]. Although these studies involved different plant types and parts, a close relationship between the changes in the organic acid amount and sugars was reported. On the other hand, there are several reports that the amount of nucleotides in plants changes in response to phosphate or oxygen deficiency [49,50,51,52]. In addition, the glycolytic system, which involves many reactions with nucleotides such as ATP, responds to nutrient deficiency, osmotic stress, drought, and cold [53]. Furthermore, some nucleotide compounds, such as cAMP or ppGpp, are thought to act as signal response factors in plants [54,55,56], and the dynamics of these signal factors may change in response to changes in climate conditions. Further studies are needed to investigate whether the nucleotides that fluctuated in this study could serve as biomarkers for differences in climate conditions before the harvest date.

In this study, we clarified the effects of sunlight duration and the average temperature on edamame components by comparing the meteorological conditions in the days before harvest and the edamame metabolome over a 6-year period. The amount of edamame components varied greatly depending on the harvest date, including the amino acids that have an important effect on edamame flavor. Although more detailed research is required, edamame harvested after a series of sunny and hot days may have a better flavor. These effects are expected to differ not only depending on the type and variety of agricultural products but also on fertilization and the original soil composition. To produce high-quality agricultural products, it may be necessary to use climate conditions before the harvest date as an indicator.

## 4. Materials and Methods

### 4.1. Harvesting and Sampling of Edamame

Edamame ‘Shirayama’ was grown in a commercial field at Terada, Tsuruoka city, Yamagata prefecture, Japan (38°43′28″ N, 139°48′03″ E), without shading and was watered appropriately. Edamame’s growth was tightly monitored, and it was harvested in the morning (7 a.m. to 8 a.m.). The harvest period was from 11 to 28 August in 2013, from 13 to 30 August in 2014, from 13 August to 6 September in 2015, from 11 to 28 August in 2016, from 18 August to 4 September in 2017, and from 13 to 27 August in 2018. Edamame pods were boiled by soaking them in boiling water for 3 min. The fresh weight of five or six seeds of edamame was weighed, then they were freeze-dried for 2 days, crushed using a mortar and pestle, and the weight of the dry powder was weighed (dry weight). Sampling was done by taking triplicates.

### 4.2. Collection of Climate Conditions

The air temperature was measured with a data logger (TR-74Ui, TANDD, Nagano, Japan) installed near the field. Tsuruoka City’s sunshine duration, rainfall, wind speed, and temperature difference between the highest and the lowest temperature in a day were obtained from the Japan Meteorological Agency’s database (https://www.data.jma.go.jp/stats/etrn/index.php, accessed on 18 March 2019, in Japanese). The sunshine duration and rainfall were measured in h in a day. An accumulated sunshine duration means the sum of the sunshine duration (h) several days before harvest. For example, SD_3 at 15 August is the sum of the sunshine duration for three days: 12, 13, and 14 August.

### 4.3. Sample Extraction

Each sample (10 mg DW) was extracted using a vortex with 500 µL of methanol containing 8 μM of two reference compounds (methionine sulfone for the cation and camphor 10-sulfonic acid for the anion analyses), 500 μL of chloroform, and 200 μL of water. The extracted solution was centrifuged at 15,000 g for 3 min at 4 °C (MX-305, TOMY, Tokyo, Japan). The upper layer was evaporated for 30 min at 45 °C by a centrifugal concentrator (Thermo Fisher Scientific K.K., Tokyo, Japan) to obtain two layers. For removing high-molecular-weight compounds, such as oligo-sugars, the upper layer was centrifugally filtered through a PALL Nanosep 3-kDa cutoff filter at 9100 g for 90 min at 4 °C. The filtrate was dried for 120 min by a centrifugal concentrator. The residue was dissolved into 20 μL of water containing 200 μM of internal standards (3-aminopyrrolidine for cation and trimesic acid for anion analyses) that were used for the compensation of migration time in the peak annotation step.

### 4.4. CE-MS Conditions

All CE-MS experiments were performed using an Agilent G7100A CE Instrument (Agilent Technologies, Sacramento, CA, USA), an Agilent G6224A TOF LC/MS system, an Agilent 1200 Infinity series G1311C Quad Pump VL, and the G1603A Agilent CE-MS adapter and G1607A Agilent CE-ESI-MS sprayer kit. The G1601BA 3D-CE ChemStation software version B.04.03 for CE and G3335-64002 MH Workstation were used. Separations were carried out using a fused silica capillary (50 μm i.d. × 100 cm total length) filled with 1 M formic acid for the cation analyses or with 20 mM ammonium formate (pH 10.0) for the anion analyses as the electrolyte. The capillary temperature was maintained at 20 °C. The sample solutions were injected at 50 mbar for 15 sec (15 nl). The sample tray was cooled below 10 °C. Prior to each run, the capillary was flushed with electrolytes for 5 min. The applied voltage for separation was set at 30 kV. Fifty percent (*v*/*v*) methanol/water containing 0.5 μM reserpine was delivered as the sheath liquid at 10 μL/min. ESI-TOFMS was conducted in the positive ion mode for the cation analyses or in the negative ion mode for the anion analyses, and the capillary voltage was set at 30 kV. A flow rate of heated, dry nitrogen gas (heater temperature 300 °C) was maintained at 10 l/min. The fragmentor, skimmer, and Oct RFV voltage were automatically set at optimum values. Automatic recalibration of each acquired spectrum was performed using the reference masses of reference standards. The methanol dimer ion ([2M + H]^+^, *m*/*z* = 65.0597) and reserpine ([M + H]^+^, *m*/*z* = 609.2806) for the cation analyses, or the formic acid dimer ion ([2M − H]^−^, *m*/*z* = 91.0037) and reserpine ([M − H]^−^, *m/z* = 607.2661) for the anion analyses provided the lock mass for exact mass measurements. Exact mass data were acquired at a rate of 1.5 cycles/sec over a 50–1000 *m*/*z* range. In every single sequence analysis (maximum 36 samples) on our CE-TOF-MS system, we analyzed the standard compound mixture at the first and the end of the sample analyses. The detected peak area of the standard compound mixture was checked at the point of reproducible sensitivity. A standard compound mixture, composed of major detectable metabolites, including amino acids and organic acids, was newly prepared at least once every half year. In all analyses in this study, there were no differences in the sensitivity of the mixture of standard compounds.

### 4.5. Statistical Analysis

Data describing the climate conditions and the quantified amounts of metabolites were integrated into one sheet. These data were standardized by subtraction of the averages from each amount and division of the resulting values by the standard deviations. The standardized data were subjected to Metaboanalyst 5.0 (https://www.metaboanalyst.ca/, accessed on 18 March 2019). For the correlation heatmap, we used Pearson’s correlation coefficient to measure distances. Euclidean distance and the Ward method were used for the cluster analysis.

## Figures and Tables

**Figure 1 plants-13-00087-f001:**
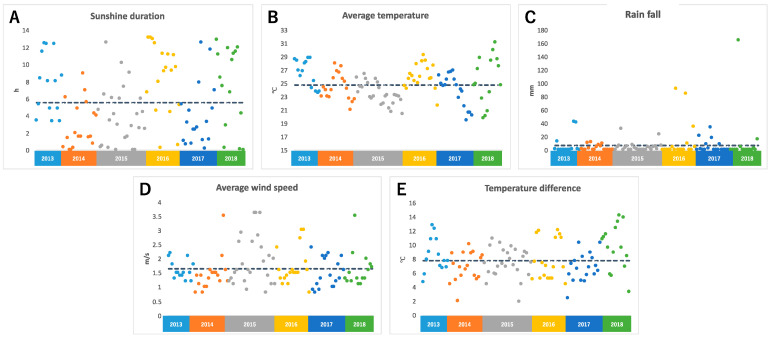
Climate conditions before harvest day: (**A**) sunshine duration, (**B**) average temperature, (**C**) rain fall, (**D**) average wind speed, (**E**) temperature difference. Each color represents data for each year. The dotted line shows the average values of each climate condition for 6 years.

**Figure 2 plants-13-00087-f002:**
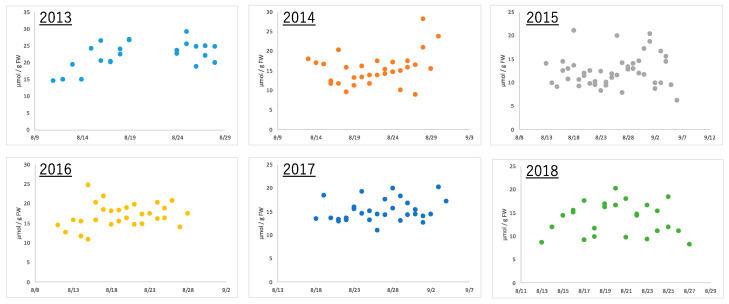
The amount of alanine in edamame at each harvest day for 6 years.

**Figure 3 plants-13-00087-f003:**
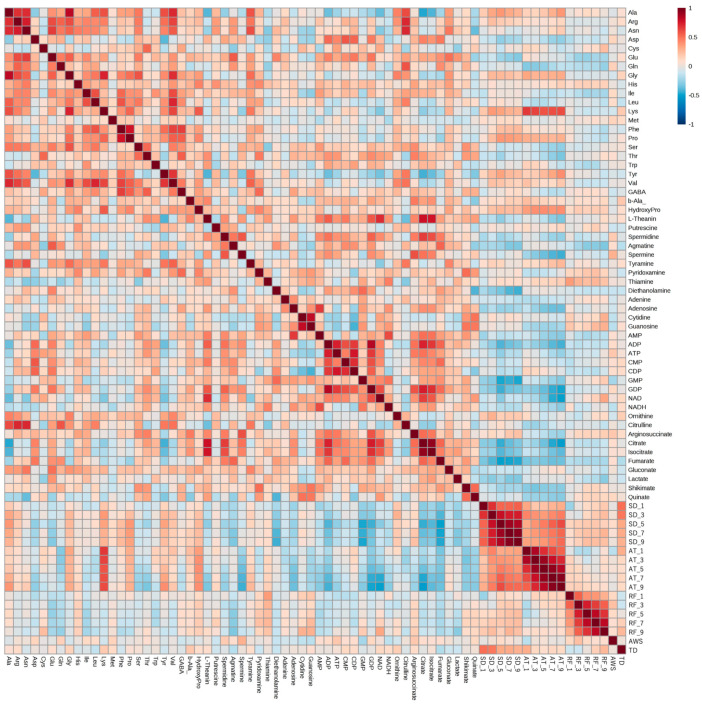
Correlation analysis between climate conditions and edamame metabolome. The color scale is shown at the top right of the figure. Red and blue colors represent higher and lower amounts of each compound and values of each climate condition, respectively. SD: sunshine duration; AT: average temperature; RF: rainfall; AWS: average wind speed; TD: temperature difference. SD_x means the SD amount accumulated x days before harvest.

**Figure 4 plants-13-00087-f004:**
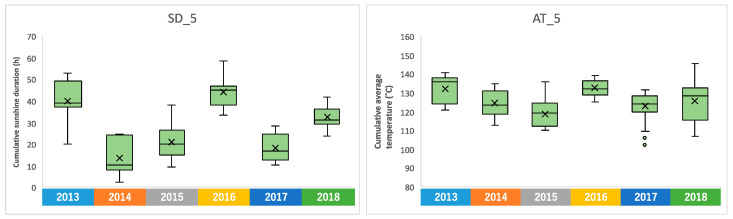
Annual fluctuation of the accumulated sunshine duration and average temperature during the 5 days before harvesting, shown as SD_5 and AT_5, respectively.

**Figure 5 plants-13-00087-f005:**
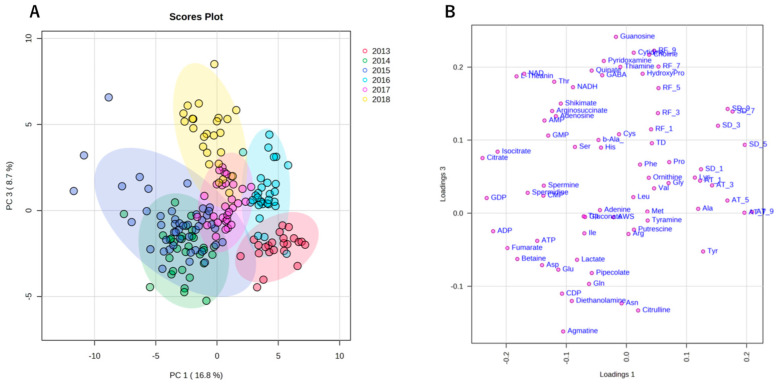
Principal component analysis (PCA) using climate conditions and edamame metabolome for 6 years. Each color indicates data for each harvest year. The scores plot showed the cluster distribution of the harvest years along both the PC1 and PC3 axes (**A**). The loading plot shows climate conditions or compounds with significant influence on clustering (**B**). SD: sunshine duration; AT: average temperature; RF: rainfall; AWS: average wind speed; TD: temperature difference. SD_x means the SD amount accumulated x days before harvest.

**Figure 6 plants-13-00087-f006:**
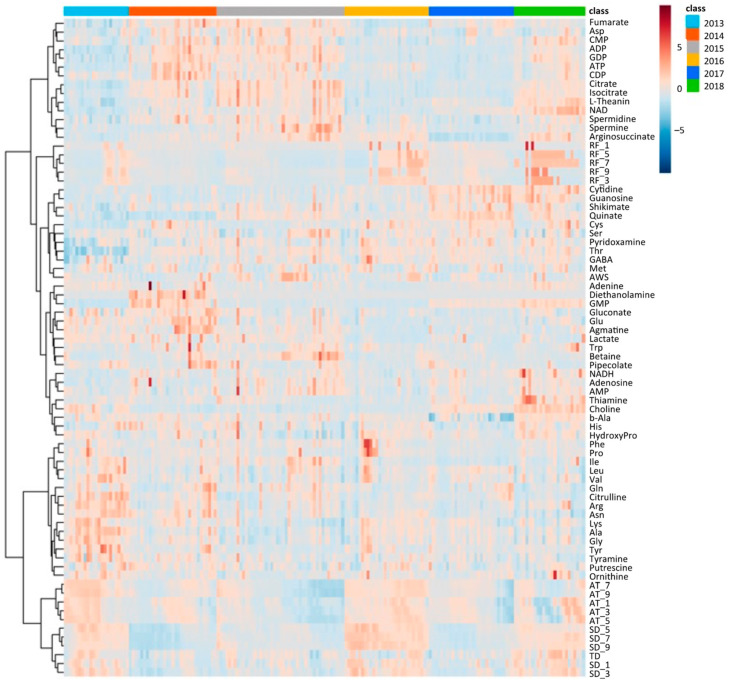
Hierarchical cluster analysis using climate conditions and edamame metabolome for 6 years. Color bar shows harvest year (top right, right), and color scale shows amounts of each compound or values of each climate condition (top right, left); red and blue colors mean higher and lower values. SD: sunshine duration; AT: average temperature; RF: rainfall; AWS: average wind speed; TD: temperature difference. SD_x means the SD amount accumulated x days before harvest.

**Figure 7 plants-13-00087-f007:**
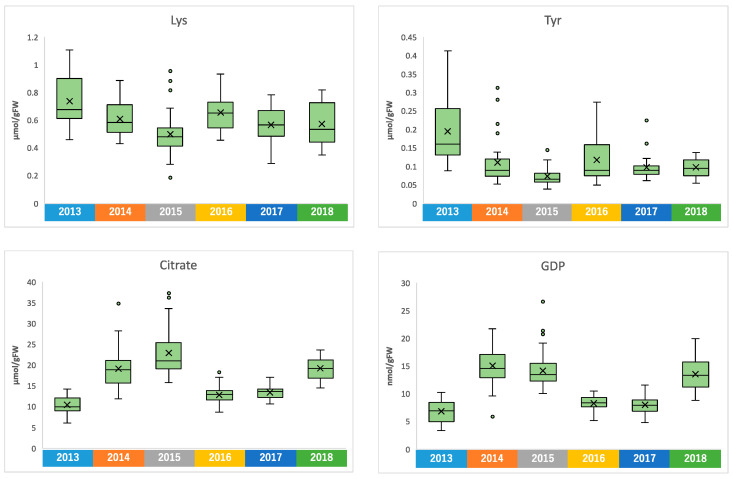
Amounts of Lys, Tyr, citrate, and GDP in each harvest year.

**Table 1 plants-13-00087-t001:** Top five compounds that showed a correlation or inverse correlation with sunshine duration (SD). The ranking was determined by the sum of the correlation coefficients from SD_1 to SD_9. Bold numbers indicate the highest correlation or inverse correlation coefficient between SD_1 and SD_9. SD_x means the SD amount accumulated x days before harvest.

Compounds	SD_1	SD_3	SD_5	SD_7	SD_9
Lys	0.20	**0.37**	0.34	0.28	0.27
Ala	0.18	0.32	**0.38**	0.31	0.27
Pro	0.05	0.23	**0.38**	0.36	0.34
Tyr	0.13	0.25	**0.34**	0.25	0.23
Hydroxy Pro	0.10	0.22	**0.26**	0.24	0.24
Fumarate	−0.33	−0.40	**−0.49**	−0.48	−0.47
GMP	−0.26	−0.30	−0.47	−0.44	**−0.48**
Citrate	−0.28	−0.30	**−0.44**	−0.36	−0.31
GDP	−0.24	−0.29	**−0.43**	−0.35	−0.33
Diethanolamine	−0.20	−0.28	−0.33	−0.40	**−0.43**

**Table 2 plants-13-00087-t002:** Top five compounds that showed a correlation or inverse correlation with average temperature (AT). The ranking was determined by the sum of the correlation coefficients from AT_1 to AT_9. Bold numbers indicate the highest correlation or inverse correlation coefficient between AT_1 and AT_9. AT_x means the AT amount accumulated x days before harvest.

Compounds	AT_1	AT_3	AT_5	AT_7	AT_9
Lys	**0.67**	0.64	0.59	0.60	0.61
Hydroxy Pro	0.35	0.40	**0.41**	0.39	**0.41**
Gly	**0.33**	0.28	0.27	0.30	0.31
Pro	**0.30**	0.29	0.27	0.27	**0.30**
Tyr	0.16	0.17	0.25	0.33	**0.41**
GDP	−0.27	−0.30	−0.36	−0.45	**−0.48**
Citrate	−0.19	−0.24	−0.35	−0.47	**−0.51**
Spermine	−0.23	−0.29	−0.36	−0.41	**−0.43**
Fumarate	−0.33	−0.30	−0.33	**−0.36**	−0.35
NAD	−0.16	−0.20	−0.32	−0.46	**−0.50**

## Data Availability

The metabolome data used in this study will be made available on the public repository MetaboBank (https://www.ddbj.nig.ac.jp/metabobank/index-e.html, accessed on 1 October 2023).

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
