# Peer review of "Effects of Climate Conditions before Harvest Date on Edamame Metabolome"

_plants, 2023, doi:10.3390/plants13010087_

Round 1
Reviewer 1 Report
Comments and Suggestions for Authors
The study reported the ‘effects of weather conditions before harvest date on edamame metabolome’. The study takes a long time and authors did a lot of work. However, it seems that the most valuable point of this study was not pointed. I think readers may much care about that harvesting the edamame under or before which kind of weather will get the best quality of Edamame, rather than just the ‘effects of weather conditions on metabolome’. If authors can give out the suggestion on harvesting the edamame under or before which kind of weather, the valuable of the study will be much better.
Author Response
Thank you very much for taking the time to review this manuscript. Following your advice, I added the following sentence to the final paragraph of Discussion section; “Although more detailed research is required, edamame harvested after a series of sunny and hot days may have a better flavor.”
Reviewer 2 Report
Comments and Suggestions for Authors
Introduction: has a very list-like quality; could stand to be expanded out and given more context rather than a long list of evidence.
Results: what are the ideal conditions for growing edamame? What were the average conditions for each category? It’s difficult to tell based on Figure 1. How much impact does the large amount of variation have on the metabolome data?
Discussion: much of the opening paragraph feels like it would be more suited to the introduction. The results of this study are not mentioned until the final third of the paragraph. Similarly, the second paragraph contains a great deal of background information and only references the study performed at the end.
[63-65] Define what is meant by “the short period before the harvest date”
[114-115] How does boiling impact the metabolome?
[163] Remove “tables maybe have a footer”
[187-189] Expand on these results in addition to directing readers to figure 6. Which were the most significant? Which were the most interesting?
Author Response
> Thank you very much for taking the time to review this manuscript. Please find the detailed responses below and the corresponding revisions highlighted changes in the re-submitted files.
Introduction: has a very list-like quality; could stand to be expanded out and given more context rather than a long list of evidence.
> I have expanded the background by adding the details of cited papers. Added sentences were highlighted at Introduction section.
Results: what are the ideal conditions for growing edamame? What were the average conditions for each category? It’s difficult to tell based on Figure 1. How much impact does the large amount of variation have on the metabolome data?
> Our point in Figure 1 is that these parameters are highly variable from day to day. To make it easier to understand, we've added a sentence about the climate conditions which can lead to high quality edamame production in the last paragraph of the Discussion.
Discussion: much of the opening paragraph feels like it would be more suited to the introduction. The results of this study are not mentioned until the final third of the paragraph. Similarly, the second paragraph contains a great deal of background information and only references the study performed at the end.
> I have added some sentences to the discussion to make the connection between the results and the discussion clear.
[63-65] Define what is meant by “the short period before the harvest date”
> To make this sentence clear, I changed this part to “for several days before harvesting”.
[114-115] How does boiling impact the metabolome?
> I added the following sentences; “Boiling the edamame immediately after harvest prevented the loss of freshness and made the edamame ready for consumption. After each sample of these boiled edamame was extracted and pre-treated using conventional methods, metabolome analysis was performed using CE-MS.”.
[163] Remove “tables maybe have a footer”
> I deleted this sentence.
[187-189] Expand on these results in addition to directing readers to figure 6. Which were the most significant? Which were the most interesting?
> I added several sentences to section 2.4.2.
Reviewer 3 Report
Comments and Suggestions for Authors
Review comments to submitted manuscript (MS).
“Effects of Weather Conditions before Harvest Date on Edamame Metabolome”
By:
Akira Oikawa, Katsutaka Takeuchi, Kei Morita, Yamato Horibe, Ryosuke Sasaki, and Hideki Murayama.
Submitted to Plants.
The MS documents the contents of edamame seeds (immature soybean) harvested under field conditions and correlates the data to the weather conditions experienced by the crop up to 9 days before the harvest.
On one hand, the work seem to have been performed thoroughly, but the results reflect previously known plant metabolic responses to environmental conditions. In this sense the contribution is not really novel.
On the other hand, the combination of natural variations seem rather complex and the authors finding may offer hints to agricultural crop management. In this sense, it could be of interest to readers of Plants.
Unfortunately, the MS is unsuitable for publication at this point, as it has some deficiencies demanding attention:
-
Data are not presented with enough clarity and care for details.
-
The discussion jumps to conclusions with little or no support.
-
Language style requires improvement.
Important issues:
0. The title should be changed “Effects of Climate Conditions before Harvest Date on Edamame Metabolome”
for the reason behind read the definitions of weather (https://public.wmo.int/en/our-mandate/weather) and climate (https://public.wmo.int/en/our-mandate/climate) at the World Meteorological Organization
1. There are problems with undefined abbreviations.
-
The abbreviation TD is found in the list of variables to the left of figure 5 and in the left panel of figure S6, yet, it goes completely undefined.
-
In tables 1 & 2 and in figures 5 and S3. The authors use the abbreviations SD_1, SD_3, SD_5, SD_7 and SD_9. This abbreviations are not defined anywhere in the text. These seem to refer to sunshine duration (as defined by the World Meteorological Organization), and the reader may guess these to be associated to the definitions given in the legend to figure S1, where the number may relate to days before harvest.
-
The same goes for AT_1 to AT_9 in table 2, figures 4, 5 and S4. and for P_1 to P_9 in figure 5 and S5.
-
All abbreviations must be defined in the main text, and when appropriate, in the corresponding legends to figures and table footnotes (or headings).
-
The abbreviation “P” (precipitation) is inconvenient, for it may be confused with the symbol of the chemical element phosphorous. Furthermore, accumulation of water by climate phenomena, can be measured as precipitation and rainfall and both terms have precise definitions. Precipitation includes snow and heil, while rainfall included only the fall of liquid water droplets. The authors probably meant rainfall (my guess), but I must not be guessing, the MS should be crystal clear.
2. The authors have some problems in the handling of important concepts, so either the terms are described incorrectly or their meaning is no fully explained.
-
For a difference you need two values (at least). But in figure 1, the temperature “difference” is reported with no indication of what was the reference to calculate such difference. Some clarification is needed.
-
The authors need to revise carefully the use and meaning of “amount” vs. “content”. Amount refers just to a certain quantity of some stuff (usually matter). Content means some amount enclosed in a defined “container”, in this case the Edamame seed. Amount does not imply a defined container, but content does.
-
The authors use “accumulated sunshine duration”, “sunshine duration” or “accumulated sunlight hours” indistinctly. Stick to one term. Then, accumulated sunshine duration is measure in h in a day or h per year. But the units in figure S1 are h and there are several dots in one year for SD 3 days. First, I have to assume these are at three days before harvest, but I cannot be sure, it could equally mean 3 day average. Second I may assume that each dot correspond to one harvesting season, but in section 4.2, each year has one harvesting season. Third I guess these could be observations from different locations, methods, or instrumentations, but the URL given as a source only give the sunshine duration in h by month. So their sunshine duration data are a mystery to me, which is completely unacceptable for a scientific paper.
-
The authors use accumulated precipitation and precipitation indistinctly, stick to one term. However consider using rainfall instead (see below).
3. Metabolic activity changes along daytime in many organisms, and specially in plants. The Authors took the care to perform the harvest Edamame at the same time of day, as indicated in methods. However it should aid the reader to restate the fact at the point where the metabolomic data is described (section2.2).
5. Since weather conditions in the MS do refer to a particular time span preceding harvest day, it becomes strictly necessary to state the length of the time period in figure legends and table heading (or table footnotes). Thus, “weather conditions” should be climate conditions recorded along 9 days before harvest.
6. In the last paragraph of page 5 (lines 150-153), the MS states: “These results suggest that the effects of weather conditions on the metabolism and transport of compounds contained in edamame have time differences depending on weather conditions and types of compounds.” I found hard to make sense of this statement beyond the obvious and trivial sentence “when climate changes metabolites change”. The authors should clarify their intent.
7. In lines 183 to 185 the MS states: “As a result, many amino acids, such as Lys and tyrosine (Tyr), contained in edamame were classified into the same cluster as sunshine duration, average temperature, and their cumulative values.”
How can an amino acid be placed in the same cluster as a weather condition?
This interpretation does not make since, the climate conditions represent stimuli given to the plant, while the metabolites are produced by the plant, and may be influenced by the stimuli. The statistical analysis was not performed in a sensible manner, the sentence could be flawed, or the description is pretty incomplete. Clarification is strongly needed here.
8. in lines 185-186 the MS reads: “Many organic acids and nucleotides, such as citric acid and GDP, were identified in clusters separated by sunshine durations.”
The statement literally puts sunshine duration (which should not take a plural, by the way) as the entity responsible to segregate some compound clusters. But that is not what cluster analysis indicates. Cluster analysis simple classifies variables by their tendency to covariate. It can be noticed in figure 2, how SD_1 to SD_9 shown higher correlation between them than with AT_1 to AT_9, and vice versa. Not surprisingly, cluster analysis puts SD_1 to SD_9 in one cluster and AT_1 to AT_9 in a second cluster. It is somehow surprising (though poorly discussed in the text) the robustness of the cluster hierarchy, because it did maintain such grouping despite the clear differences in the observed patterns along the years.
Besides, what I find in figure 5 is a cluster of SD_N (assuming these are sunshine durations at N days before harvest), AT_N (assuming these are average temperature at N days before harvest) and TD (undefined), and a second cluster of amino acids and polyamines (specifically F-P-I-L-V-Q-R-N-K-A-G-Y, plus citrulline, ornitine, tryptamine, and putrecine). These two goups showing a stronger association between them, than any of them with P_N (assuming these are rainfall values at N days before harvest). Then, an additional large group of metabolites (cytidine, guanosine, shikimate, quinate, Cys, Ser, pyridoxamine, Thr, GABA, Met, AWS, Adenosine, diethanolamine, GMP, gluconate, Glu, agmatine, lactate, Trp, pipecolane, NADH, adenosine, AMP, thiamine, choline, β-ala, His, hydroxyPr) are clustered in supergroup (subdivided in a number of smaller subgroups), while P_N values are in a separate group. Again, these last two groups do show a stronger association between them, than any of them with SD_N+AT_N+TD.
A third block of metabolites is clustered separate, showing weaker association to either SD_N+AT_N+TD or P_N. These metabolites may be considered as having a poor covariation linkage with the remaining groups.
9. In lines 186-188 the MS says: “The annual fluctuations of these characteristic compounds certainly showed fluctuations that showed a relationship or inverse relationship with sunshine duration and average temperature (Figure 6).”
But figure 6 does not have the information mentioned. Figure 6 are 4 series of wisker-box plots summarizing the observed distribution of variation in 4 metabolite contents (one on each panel) against the year of harvesting.
10. The discussion has a rhetorical style unworthy of scientific communication.
Line 19, “Changes in the environment can have a fatal impact on the survival of plants”. True, but beneficial conditions should predominate, or plant species would become extinct.
lines 198-199, “plants are thought to biosynthesize various metabolites to respond to harsh environments”. There is an overwhelming amount of scientific evidence, not just wishful thinking.
Line 214, “we focused on the weather conditions the day before harvest”. Yet the data and analysis go up to 9 days before harvesting.
Lines 219- 221 Our findings provide a new perspective on the supply of agricultural products and ways to improve flavor and health functionality. All data presented reveal the unknown details of relationships expected to exist. I do not see how this perspective is new. Furthermore, unless climate conditions are controlled, which is expensive, the data, as given, seem of little use to the farmer in the field.
Lines 235-237, “This information will be valuable in the development of future agricultural techniques such as breeding and fertilization.” The data do not deal with these two factors. It is unclear how the presented evidence could be extrapolated so far.
Lines 244-246, “Pro is thought to respond to stress as an osmolyte, and in this study, Pro content in edamame beans increased due to drought stress caused by increases in sunlight duration and average temperature.” Cluster analysis and correlations DO NOT SUPPORT CAUSAL RELATIONSHIPS. This is an unsupported statement.
Lines 251-253, “increased hydroxyPro in edamame under long sunshine duration or high-temperature conditions may be related to hydroxyPro-rich glycoproteins in edamame.” The statement is speculative and poorly documented. Proline hydroxylation in proteins does occur in the ER and its produced in situ by posttranslational protein modification (Arrigoni et al. J. Plant Physiol. (2000) 157: 481—488, Doi: 10.1016/S0176-1617(00)80102-9 ). Furthermore, the authors assume this to be the result of adaptive stress responses, but neither is there a control treatment as reference to make the plant stress evident, nor there are physiological measurements of known sings of adaptive stress responses, such as anthocyanin accumulation, stomatal closure, wilting and so on.
Lines 255-258, “since these amino acids serve as raw materials for various secondary metabolites, they may be biosynthesized as part of edamame's defense response to stress caused by sunlight and temperature.” This is again speculation with little grounds. The authors do assume the plants to be stressed, without further evidence.
Lines 261-262, “It was speculated that this was due to the high temperature.” This statement seems out of place or is incomplete. It was speculated where, when, by whom?
Lines 269-273, “Although we were not able to examine sugar content in this study, the sugar content of edamame seeds may change as the temperature rises due to increased sunlight duration. By examining the sugar content, which is a factor related to the flavor of edamame, it will be possible to investigate in more detail the influence of pre-harvest weather conditions on edamame quality.” What is the use of discussing the value of determinations you did not make. The authors should focus in discussing the results presented. The real question here is: If sugar content is so important, why it was not measured?
Lines 273-274, “On the other hand, no reports were found on the relationship between nucleotide content and abiotic stress”. The authors should review the literature. While the data may not be exhaustive, and the techniques may have been unsophisticated, there is good deal of data on adenylate pools in plant responses to several stress conditions:
- Raymond & Pradet (1980) Biochem. J. 190, 39–44, Doi: 10.1042/bj1900039
- Drew et al. Planta (1985) 165: 51--58, Doi: /10.1007/BF00392211.
- Duff et al. Plant Physiol (1989) 90: 1275-8, Doi: 10.1104/pp.90.4.1275.
- Plaxton et al. Plant Physiol (2011) 156: 1006-15 , Doi: 10.1104/pp.111.175281,
- Plaxton, Annu Rev Plant Physiol Plant Mol Biol (1996) 47: 185-214 , Doi: 10.1146/annurev.arplant.47.1.185.
11. Provide more details about the cultivation of edamames:
Was it grown with or without shading?
Were the plants watered or were fully dependent on natural rainfall?
Was this a commercial or an experimental facility? (i.e. how tight was the growth control?)
Any other details required to replicate the experiments.
12. Statistical analysis. The program used is not enough, indicate the kind of test performed, and the level of significance used.

The problems with the use of language are indicated along with other conceptual problems in the document uploaded above.
The biggest issue is the use of trivial statements and a rhetorical style.
Author Response
Thank you for your kind review. My response to your comments are shown in the attached file. Please confirm that file.

Reviewer 4 Report
Comments and Suggestions for Authors
This study examines the influence of weather conditions on Edamame metabolites, indicating the presence of multivariate data.
To enhance the scientific rigor of this study, the incorporation of multivariate data analysis techniques, such as Principal Component Analysis (PCA) and Partial Least Squares (PLS), alongside cluster analysis, is imperative.
Principal Component Analysis (PCA) plays a crucial role in identifying patterns of data clustering and detecting outliers. On the other hand, Partial Least Squares (PLS)-based methods are particularly pertinent in assessing correlations between variables.
Furthermore, there is no discussion regarding the potential presence of significant isoflavones in the samples. Are alternative assessment methods employed in addition to CE-MS? Please provide feedback in the manuscript.
Author Response
> Thank you very much for taking the time to review this manuscript. Please find the detailed responses below and the corresponding revisions highlighted changes in the re-submitted files.
This study examines the influence of weather conditions on Edamame metabolites, indicating the presence of multivariate data.
To enhance the scientific rigor of this study, the incorporation of multivariate data analysis techniques, such as Principal Component Analysis (PCA) and Partial Least Squares (PLS), alongside cluster analysis, is imperative.
Principal Component Analysis (PCA) plays a crucial role in identifying patterns of data clustering and detecting outliers. On the other hand, Partial Least Squares (PLS)-based methods are particularly pertinent in assessing correlations between variables.
> I performed PCA, however, no clear differences among harvesting years were shown. I added some sentences at section 2.4.2. and Supplementary Figure 7 for PCA.
Furthermore, there is no discussion regarding the potential presence of significant isoflavones in the samples. Are alternative assessment methods employed in addition to CE-MS? Please provide feedback in the manuscript.
> In this study, we performed metabolomics only by using CE-MS. Therefore, any neutral compounds such as isoflavones or sugars did not detected. Although we are aware that these compounds are important factors for edamame (soybean) quality, in this paper we focused our discussion on the compounds that were actually analyzed.
Round 2
Reviewer 1 Report
Comments and Suggestions for Authors
Nice work for the revision.
Author Response
Thank you again for your kind review!
Reviewer 3 Report
Comments and Suggestions for Authors
Review comments to submitted manuscript (MS).
“Effects of Climate Conditions before Harvest Date on Edamame Metabolome”
By:
Akira Oikawa, Katsutaka Takeuchi, Kei Morita, Yamato Horibe, Ryosuke Sasaki, and Hideki Murayama.
Submitted to Plants, revised version 1.
The MS documents the metabolite contents of edamame seeds harvested in the field correlates the data to the environmental conditions experienced by the plant up to 9 days before the harvest.
The data seem to have been obtained using acceptable methods. Plants metabolic responses to environmental conditions have been the subject of many studies, so the MS data are not really novel to this respect.
Climate variations natural are stochastic in nature and it may be hard to derive defined patterns, yet the authors findings may be of interest to readers of Plants.
The authors made an effort to address several issues found in the previous version and have included principal component analysis of the data. They also tried to clarify some confusing points of the previous version.
However, the paper still tries to invoke stress as an explanation to the observed patterns of metabolite variations, while the paper offers no data whatsoever on the level of stress suffered by the plant.
For example, proline is know to increase in response to osmotic stress, but different plants accumulate proline to different levels, and the authors have no ground to compare the observed variations in proline seed content to any know reference, either from the literature o from their own data. So the discussion fall repeatedly in stating “more research is needed”.
Since authors consider stress as an important condition to explain their data, they must provide evidence of the plant being under stress. Control experiments under controlled conditions could quantify the level of irradiance needed to stress the soybean plant, and record the plant temperature, stomatal opening, water loss, and proline accumulation. These data could then be used as reference.
Furthermore, there is a good deal of quantitative data published in the literature reporting plant stress responses, some of which was cited by the authors, yet they limited to mention the general fact, but did not use the published data to make quantitative comparisons of seed contents of amino acids, GABA, and other metabolites, either per seed, per g fresh weight, or in a manner suitable to establish what level of accumulation can be taken a sign of stress.
While the present version has improved in some aspects, its style has worsen, and significant problems persist. Restructuring the paper seems still necessary. A simple restating of some sentences would not be sufficient.
PLEASE SEE THE LIST OF SPECIFIC PROBLEMS IN THE ATTACHED FILE

The spelling is correct, but there are some problems with the writing style and the choice of words.
Author Response
Thank you from the bottom of my heart for your careful review. Please find the detailed responses at the attached PDF file, and revised MS and figures.

Reviewer 4 Report
Comments and Suggestions for Authors
In my review, I emphasized the importance of highlighting multivariate data analysis as a crucial aspect. The authors briefly referenced principal component analysis (PCA) in a single sentence, however, they did not provide a detailed explanation of the methodology employed for conducting the analysis. Additionally, the utilization of Partial Least Squares (PLS) methods was not executed. The utilization of these methods is crucial in identifying correlations between climate conditions and the metabolome of edamame, as well as determining the variables accountable for such correlations (or separations in the case of discriminant analysis).
Author Response
Thank you for your additional comments. We reviewed the PCA again and found clusters of harvest years in PC1 and PC3 (Figure 5). No significant cluster was found for PC2 (Supplementray Figure 7). Following your suggestion, PLS-DA was also performed and shown in Figure 6. We have also added the VIP score to the figure. A text about these results has also been added to the main text (section 2.4.2).
Round 3
Reviewer 3 Report
Comments and Suggestions for Authors
Review comments to submitted manuscript (MS).
“Effects of Climate Conditions before Harvest Date on Edamame Metabolome”
By:
Akira Oikawa, Katsutaka Takeuchi, Kei Morita, Yamato Horibe, Ryosuke Sasaki, and Hideki Murayama. Submitted to Plants, revised version 2.
The MS details how the metabolite contents of edamame seeds harvested in the field correlates the environmental conditions experienced by the plant up to 9 days before the harvest.
The data seem to have been obtained using acceptable methods, though Plant metabolic responses to environmental conditions have been the subject of many studies, and the MS data are not really novel from this point of view.
Since climate regimes in nature are chaotic, it may be hard to isolate response patterns in plants subjected to natural uncontrolled conditions; yet, the authors used in-depth statistical analysis to support their findings and these should be of interest to readers of Plants.
Though the wealth of statistical data presented may have not been fully exploited, the reader is now offered a better opportunity to draw his own hypothesis, and this should encourage further research in the area.
I recommend publication after a few minor phrasing problems are fixed.

Still a few style issues are detected, please see the suggestion in the attached document.
Author Response
Thank you very much for your kind review. Please find the detailed responses at the attached PDF file, and revised MS and figures.

Reviewer 4 Report
Comments and Suggestions for Authors
For this instance, the authors employed PLS-DA and incorporated all six datasets into a single model. In this manner, it is not feasible to identify crucial variables that contribute to the differentiation between groups. It is preferable to compare two datasets using a single model.
Moreover, the preprocessing steps undertaken before conducting MVA on the data remain undisclosed.
Author Response
Thank you for your advice. Since this study does not compare only two data, the PLS-DA was deleted. The related text and figure (Figure 6) were excluded. Regarding data processing before MVA, it is stated in section 4.5 as follows; "These data were standardized by subtraction of the averages from each amount and division of the resulting values by the standard deviations." Neither the climate data nor the metabolome data were processed in any other way before MVA.
Round 4
Reviewer 4 Report
Comments and Suggestions for Authors
It can be accepted in the present form.